# Influence of Alloyed Ga on the Microstructure and Corrosion Properties of As-Cast Mg–5Sn Alloys

**DOI:** 10.3390/ma12223686

**Published:** 2019-11-08

**Authors:** Jing Ren, Enyu Guo, Xuejian Wang, Huijun Kang, Zongning Chen, Tongmin Wang

**Affiliations:** Key Laboratory of Solidification Control and Digital Preparation Technology (Liaoning Province), School of Materials Science and Engineering, Dalian University of Technology, Dalian 116024, China; 18840830427@163.com (J.R.); eyguo@dlut.edu.cn (E.G.); wangxuejian0618@163.com (X.W.); kanghuijun@dlut.edu.cn (H.K.)

**Keywords:** magnesium, immersion test, polarization, microstructure, corrosion resistance

## Abstract

In this paper, the microstructures and corrosion behaviors of as-cast Mg–5Sn–*x*Ga alloys with varying Ga content (*x* = 0, 0.5, 1, 2, 3 wt %) were investigated. The results indicated that Ga could not only adequately refine the grain structure of the alloys, but could also improve the corrosion resistance. The microstructures of all alloys exhibited typical dendritic morphology. No Ga-rich secondary phases were detected when 0.5 wt % Ga was added, while only the morphology of Mg_2_Sn phase was changed. However, when the addition rate of Ga exceeded 0.5 wt %, an Mg_5_Ga_2_ intermetallic compound started to form from the interdendritic region. The volume fraction of Mg_5_Ga_2_ monotonically increased with the increasing Ga addition level. Although Mg_5_Ga_2_ phase was cathode phase, its pitting sensitivity was weaker than Mg_2_Sn. In addition, the standard potential of Ga (−0.55 V) was lower than that of Sn (−0.14 V), which relieved the driving force of the secondary phases for the micro-galvanic corrosion. An optimized composition of 3 wt % Ga was concluded based on the immersion tests and polarization measurements, which recorded the best corrosion resistance.

## 1. Introduction

Magnesium alloys, with their excellent properties of low density and high specific strength, have received extensive attention in automobile, aerospace, electronics, and other industries [1,2,3,4]. Among many of the binary alloy systems, the Mg–Sn phase diagram demonstrates that the Mg-rich side has a very shallow α-Mg solvus curve, indicating that it is conducive to maximizing the precipitation of the thermally stable Mg_2_Sn particles. This infers that adding Sn potentially helps to enhance the mechanical integrity of the alloy at elevated temperatures [5,6,7,8,9]. On this account, the Mg–Sn-based alloy system is one of the most extensively studied alloys in recent years [6,7,8,9,10,11,12,13].

Aside from the mechanical strength, other factors that limit the application of magnesium alloys are their high chemical activity and poor corrosion resistance. It was documented, however, that alloying with some Sn results in superior corrosion resistance. Ha et al. [14] reported that Sn decreases the cathodic current density and inhibits the cathodic H_2_ evolution. Similar effects were also observed in Mg–5Al–1Zn alloy [11] and Mg–7Al–0.2Mn alloy [12]. The reason for inhibition of H_2_ evolution by alloying Sn was studied. Ha et al. [13] observed that the main reason for the decrease of H_2_ evolution rate was that Sn elements were enriched on the metal surface. They further reported that Sn had a higher potential H_2_ evolution than the Mg matrix [15,16]. Moreover, under the condition of a high cathode, Sn could form SnH_4_ [14]. That is to say, when the matrix is corroded, the element Sn is spontaneously enriched on the matrix surface, acting as a barrier to further corrosion [13].

The addition of Ga in magnesium alloy has important research value in seawater battery and sacrificial anode [17]. In those applications, Ga enhances the mechanical performance of Mg-based alloys by solid solution and precipitation strengthening [17,18]. It was suggested that Mg–Ga, Mg–In, and Mg–Sn alloys have potential use as biomaterials [15]. Among the aforementioned three binary alloys, Mg–Ga alloys appeared to be the best candidates, considering both the mechanical properties and corrosion behaviors in 0.9 wt % NaCl solution. Marta et al. [19] pointed out that the superior corrosion resistance of Mg–Ga is mainly related to the microgalvanic corrosion between α-Mg and the second phases, and the morphology and distribution of Mg_5_Ga_2_ phase.

The purpose of this study is to explore the effect of Ga on the microstructure and corrosion resistance of Mg–5Sn alloy. Both potentiodynamic polarization measurements and immersion weightlessness experiments were conducted to evaluate the corrosion property of the experimental Mg–5Sn–*x*Ga alloys. By investigating the microstructures of Mg–5Sn–*x*Ga alloys with varying Ga contents, the immanent relationship between the microstructure and corresponding corrosion behavior is established, and the results are discussed.

## 2. Materials and Methods

### 2.1. Specimen Preparation

As-cast Mg–5Sn–*x*Ga (*x* = 0, 0.5, 1, 2, and 3, all in weight percentage unless otherwise specified) alloys were investigated in this work. The experimental alloys were prepared by melting pure metals in an electric resistance furnace under the protection of an atmosphere containing CO_2_ and SF_6_ mixture in a ratio of 99:1. Pure Mg (99.99%) ingot was cut into small pieces and put in a magnesia crucible. Once the temperature of the melt reached 973 K, pure Sn (99.99%) and Ga (99.999%) granules were added to the melt. The melt was fully stirred with a graphite impeller for 180 s before it was poured into a cylindrical steel mold preheated at 523 K. The chemical compositions of the alloys were measured by X-ray fluorescence analysis (XRF-1800, Shimadzu, Kyoto, Japan), and the results are summarized in Table 1.

### 2.2. Microstructural Characterization

The samples for microstructural observation were cut 10 mm from the bottom of the ingot. The exposed surfaces were ground by SiC abrasive papers up to 4000 grit and then polished up to 1 μm in a suspension of diamond pastes. The polished surfaces were etched in a mixed solution of picric and acetic acid (10 mL acetic acid, 4.2 g picric acid, 10 mL distilled water, and 70 mL ethanol). The microstructures and corrosion morphologies of the specimens were observed using an optical microscope (OM, Olympus GX51, Olympus Corp., Tokyo, Japan) and a field emission scanning electron microscope (FESEM, Zeiss supra 55, Zeiss Corp., Oberkochen, Germany), equipped with energy-dispersive X-ray spectroscopy (EDS). Phases were identified by an X-ray diffraction instrument (XRD, PANalytical Empyrean, Almelo, The Netherlands) with Cu K_α_ radiation at a scanning speed of 0.142224°/s. An electron microprobe analyzer (EPMA, JXA-8530F PLUS, Tokyo, Japan) equipped with a wavelength dispersive spectrometer (WDS) was used to examine the elemental distribution of selected phases. Grain size was measured using the OM micrographs of the alloys by the intercept method regulated in GB/T 6394-2002:(1)$$\tau =\frac{L}{M\times N}$$
where *τ* is the average grain size, *L* is the measured mesh length, *M* is the magnification for observation (50× in this work), and *N* is the number of intercept points on the measured mesh as indicated in Equation (1). The grain sizes were averaged with values measured from six micrographs for each alloy.

### 2.3. Corrosion Tests

The corrosion behaviors of the as-cast Mg–5Sn–*x*Ga (*x* = 0, 0.5, 1, 2, and 3 wt %) alloys were investigated using a potentiodynamic polarization test and immersion test. All experiments were carried out at room temperature (298 ± 1 K).

For the electrochemical test, a conventional three-electrode cell was employed. The cell was composed of a working electrode (sample), a reference electrode (Ag/AgCl electrode, saturated KCl with electrode potential of 0.1981 V vs. Standard Hydrogen Electrode), and a Pt plate counter electrode. The test was carried out using a Gamry electrochemical workstation (Reference 600, Gamry Instruments Inc., Warminster, PA, USA). The samples were ground using SiC abrasive papers up to 2000 grit, and the surface of a round specimen with an area of 1.0 cm^2^ was exposed to 3.5 wt % NaCl solution. After reaching a steady open circuit potential (OCP), the polarization test was initiated from −0.3 V versus the OCP level of the sample to +0.3 V vs. OCP at a scanning rate of 1 mV/s. Three samples were tested for each alloy to ensure repeatability. The corrosion current density (*i*_corr_, mA/cm^2^) of the investigated alloys, which was obtained by fitting the cathodic branch of the potentiodynamic polarization curve, was converted to the average corrosion rate (*R_i_*, mm/y) according to the following equation [20,21]:(2)$$R_{i}=22.85\times i_{\mathrm{corr}}$$

The immersion test was performed on three parallel samples for each alloy. Samples with dimensions of 20 mm wide, 25 mm long, and 3 mm thick were mechanically ground using SiC abrasive papers up to 2000 grit. The ground samples were immersed in a 3.5 wt % NaCl solution at 298 K for 72 h. After immersion, the samples were taken out and cleaned with chromate acid (200 g/L CrO_3_ + 10 g/L AgNO_3_) to remove the corrosion products [22,23]. The obtained samples were rinsed with distilled water, cleaned in alcohol, and dried in air. The final specimens were weighed on an analytical balance (ME204E, Mettler Toledo Corp., Greifensee, Switzerland), and the corrosion rate (*C*_R_) for each sample was calculated by Equation (3) [24,25]:(3)$$C_{R}=\frac{K\times W}{A\times T\times D}$$
where *K* is a constant (8.76 × 10^4^), *W* is the mass loss in g, *A* is the exposed area of the sample in cm^2^, *T* is the exposure time in hours, and *D* is the density in g/cm^3^.

Another group of samples for observation of corrosion surface morphology, corrosion depth, and products identification were Φ 12 mm in diameter and 3 mm in thickness. These samples were prepared following the same procedure as the samples for microstructural observation mentioned in Section 2.2. Measures were taken to ensure that corrosion occurred only on the transverse surface of the cylindrical samples. After immersion in 3.5 wt % NaCl solution for 24 h, XRD was used to identify the phases in the corrosion product. The surface and cross-section morphologies of the corroded samples were observed in SEM and EPMA, respectively.

## 3. Results and Discussion

### 3.1. Microstructure Analysis

The OM micrographs of the as-cast Mg–5Sn–*x*Ga alloys with varying Ga contents are shown in Figure 1. All the alloys present a similar typical dendritic grain structure. The solute elements could have significant effects on the growth behavior of α-Mg grains and the final morphological patterns [26,27,28,29]. It was documented that Sn could form a composition undercooling zone of liquid ahead of the solid–liquid interface, leading to dendrite formation [30]. The average grain sizes were measured to be 300.4 ± 18.4, 227.1 ± 19.3, 205.7 ± 16.7, 183.3 ± 12.7, and 155.9 ± 8.6 μm for the Mg–5Sn, Mg–5Sn–0.5Ga, Mg–5Sn–1Ga, Mg–5Sn–2Ga, and Mg–5Sn–3Ga alloys, respectively. The result suggests that the grain size decreased monotonically with increasing Ga content, probably due to the increasing grain growth restriction factor (*GRF*), which can be expressed by Equation (4) for a binary alloy [26]:(4)$$GRF=m_{L}C_{0}\left(k_{0}-1\right)$$
where *m*_L_ is the slope of liquidus (assumed to be straight), *k*_0_ the equilibrium distribution coefficient, and *C*_0_ the concentration of a solute. According to the Mg–Ga phase diagram [31], in dilute binary magnesium alloys, the *GRF* value for Ga element is calculated to be 4.04, which is similar to Al and Zn element (*GRF*_Al_ = 4.32, *GRF*_Zn_ = 5.31 [26]). Composition undercooling was established by the Ga concentration gradient in the diffusion layer adjacent to the solid–liquid interface, restricting grain growth as a consequence of slow diffusion; and thus, the growth rate is limited and grain size is refined.

As for the case of the Mg–5Sn–0.5Ga alloy, the quantitative results from the WDS analysis of the areas marked by the white rectangles in Figure 1f are given in Table 2. It was found that Mg–5Sn alloy with 0.5 wt % Ga did not cause the formation of any new phases. Most of the Ga and a part of the Sn were either dissolved in the α-Mg matrix or enriched at the interdendritic region (indicated by arrow *a*).

Dozens of studies have shown that the microstructural features, such as the grain size and the morphology of second phase, may have impacts on the corrosion resistance of magnesium alloys [32,33]. Grain refinement led to the decreased corrosion rates, possibly because of the enhanced passivity of the oxide film [34]. Because the experimental alloy with 3 wt % Ga had the finest equiaxed grains, it was most likely that the as-cast Mg–5Sn–3Ga alloy would exhibit the greatest corrosion resistance.

The backscattered electron (BSE) micrographs of the Mg–5Sn–*x*Ga alloys with varying Ga content are shown in Figure 2a–e. The OM and BSE micrographs of the alloys confirm the presence of second phases and element enrichment in the interdendritic regions. The elemental mapping results of the Ga, Mg, and Sn in the Mg–5Sn–3Ga sample using EPMA are shown in Figure 2f–i. Contrasts along the interdendritic contours demonstrate that there are two types of phases with different chemistries. According to the quantitative results measured by WDS in Table 3, the grey eutectic phase indicated by the letter *a* is Mg_5_Ga_2_ intermetallic compound, and the brighter phase indicated by the letter *b* is Mg_2_Sn. The inserts in Figure 2a–e are zoom-in views of the two phases embedded in the interdendritic region. When 0.5 wt % Ga was added, no Ga-rich second phases were detected, according to the composition analysis in Table 2. When the addition of Ga exceeded 0.5 wt %, Ga formed from the interdendritic region in the form of Mg_5_Ga_2_ phase, while Mg_2_Sn did so in the form of divorced eutectic phase.

The XRD spectra in Figure 3a further confirmed that the Mg–5Sn alloy consisted of α-Mg matrix and Mg_2_Sn. When the Ga content exceeded 1 wt %, a new phase containing Ga was formed. Survey on Joint Committee on Powder Diffraction Standards indicates that the new peaks correspond to the reflections of the Mg_5_Ga_2_ compound, which normally exists in Mg–Ga binary [15,35] and Mg–Hg–Ga ternary [18] alloys. This is consistent with the quantitative results measured by WDS in Table 3. The increasing density of the Mg_5_Ga_2_ peaks is expected with the increase in the Ga content of the alloys. Figure 3b shows the area fraction of the second phases in the investigated alloy calculated by ImageJ 1.47 (US NIH, Bethesda, MD, USA) [36]. It is also clear that the area fraction of Mg_5_Ga_2_ particles increased with increasing Ga content, while the area fraction of Mg_2_Sn particles decreased accordingly.

### 3.2. Polarization Tests

Figure 4 shows the potentiodynamic polarization curves of the as-cast Mg–5Sn–*x*Ga alloys. The corrosion current density (*i*_corr_), corrosion potential (*E*_corr_), and the average corrosion rate (*R*_i_) obtained from the potentiodynamic polarization curves are listed in Table 4. The *i*_corr_ of the as-cast Mg–5Sn–*x*Ga alloys in 3.5 wt % NaCl solution increase in the order of Mg–5Sn–3Ga, Mg–5Sn–2Ga, Mg–5Sn–0.5Ga, Mg–5Sn–1Ga, and Mg–5Sn. The lowest corrosion current density and corrosion potential are 2.79 × 10^−2^ mA/cm^2^ and −1.684 V for Mg–5Sn–3Ga alloy, respectively.

It is accepted that the corrosion of Mg-based alloys is generally caused by the cathodic reaction of hydrogen evolution. The anodic reaction of Mg dissolution can be described as [37,38]:
Anodic reaction: Mg → Mg^+^ + e^−^(5)
Mg^+^ + H_2_O → Mg^2+^ + OH^−^ + ½H_2_(6)
Cathodic reaction: 2H_2_O + 2e^−^ → H_2_ + 2OH^−^(7)

Ga is reported to have relatively high hydrogen overpotentials [15]. Alloys with Ga are thus anticipated to inhibit the cathodic reaction of Equation (6). The cathodic branches of the polarization curves show that the cathodic current density decreases with increasing Ga content, demonstrating that Ga indeed delays cathodic reaction and thus reduces corrosion rate.

Figure 5a shows the representative anode branches of the polarization curves of the Mg–5Sn–*x*Ga alloys in 0.01 M NaCl solution. Within the whole range of anode polarization, the relation between the passive current density (*i*_passive_) of the alloys at any potential remains unchanged, so the relation between the current density of the alloys at an eigenvalue potential can be selected to describe the change of the trend of the passive current density of the alloys after adding Ga element. This method is favored by scholars in many studies. Ha et al. [39,40] used this method to study the change trend of anode passive current density at −1.7 V and cathode hydrogen evolution current density at −1.9 V of Mg–5Sn–(1–4 wt %)Zn alloy system. Similarly, Kim et al. [41] studied the polarization curve of Mg–8Sn–1Zn alloy with this method. In order to explore the role that Ga played in the anodic passive layer, Figure 5b compares the passive current density values for the alloys of different Ga contents with the alloy without Ga addition. The *i*_passive_ values were measured at the anodic potentials of −1.65 V. Evidently, compared with the Mg–5Sn alloy, the *i*_passive_ values decreased after adding Ga element, among which the Mg–5Sn–0.5Ga alloy exhibited the lowest *i*_passive_ value. Decreased *i*_passive_ indicates enhanced passive film stability. As mentioned above, Mg_5_Ga_2_ phase will be formed when Ga addition exceeds 0.5 wt %, which belongs to the cathode phase and destroys the stability of passive film to some extent. Moreover, the passive film of magnesium alloy is extremely unstable and cannot protect the matrix, so the corrosion rate is mainly controlled by hydrogen evolution reaction of the cathode.

Another factor that also influences the corrosion rate is the microstructural features of an alloy [42,43,44]. With the increase in Ga element, the area fraction of Mg_2_Sn particles monotonically decreases, while the opposite trend is the case for the Mg_5_Ga_2_, as shown in Figure 2 and Figure 3. Liu et al. [16] reported that both the hydrogen evolution rate and corrosion potential decreased with decreasing volume fraction of the Mg_2_Sn particles, which is consistent with the experimental results in this study.

### 3.3. The Immersion Tests

Figure 6 shows the average corrosion rates of the alloys with continuously increasing Ga content calculated by Equation (2). In essence, the variation trend of the corrosion rate obtained by immersion test is in accord with the potentiodynamic polarization results. The lowest weight loss rate is 1.15 ± 0.1 mm/y for the Mg–5Sn–3Ga alloy. This confirms that the addition of Ga improves the corrosion resistance of Mg–5Sn alloy.

After immersion at 298 K for 6 h, the corrosion initiation sites were observed using SEM and EDS. Figure 7 presents the representative BSE micrographs of three alloys (Mg–5Sn, Mg–5Sn–0.5Ga, and Mg–5Sn–3Ga) and the corresponding EDS spectra of the marked areas. Figure 7a,b reveals that the Mg_2_Sn phase was not covered with corrosion films, and cracks were formed in the vicinity of Mg_2_Sn phase. Combining with the EDS analysis of points *A* and *B* (Figure 7d,e), it is inferred that corrosion began in the magnesium matrix around the Mg_2_Sn phase. Moreover, the adjacent distribution of the two types of phases provides the possibility for analyzing the initial corrosion location. In the Mg–5Sn–3Ga alloy (Figure 7c), the phase at point *C* is rich in Ga element compared with its neighboring phase (marked by point *D*), suggesting that points *C* and *D* are Mg_5_Ga_2_ phase and Mg_2_Sn phase, respectively. Also shown in Figure 7c is that the surface film on Mg_2_Sn phase was broken, while this was not the case for the film on Mg_5_Ga_2_ phase. Mg_2_Sn phase was detached from the matrix after immersion for 6 h. All the aforementioned observations indicate that Mg_2_Sn phase, rather than the Mg_5_Ga_2_ phase, provided the sites to initiate corrosion. Moreover, compared with the larger Mg_2_Sn phase, eutectic Mg_5_Ga_2_ phase is deemed to stay more firmly in the alloy matrix. Therefore, with the increase in Ga content, the increasing area fraction of Mg_5_Ga_2_ further improved the corrosion resistance of the alloys.

Figure 8 shows the evolution of the surface morphologies of the investigated alloys after immersing for different durations. The surfaces of the Mg–5Sn–*x*Ga alloys are covered by a mixture of dark film and bright corrosion product. A number of deep corrosion pits are observed after immersion (Figure 8a). It is noticeable that the white corrosion products increase with the decrease in Ga content.

It is noted in Figure 8 that the Mg–5Sn–0.5Ga alloy was more seriously corroded than the Mg–5Sn–1Ga alloy, while the immersion tests produced an opposite result. The reason is that in the Mg–5Sn–0.5Ga alloy, there is no Ga-rich second phase, which has a marginal effect on initiating corrosion. In addition, Sn and Ga elements are more passive than Mg (the standard potentials of Mg, Sn, and Ga are −2.36 V, −0.14 V, and −0.55 V, respectively) [45]. The dissolved Ga element was expected to increase the potential of α-Mg, reducing the potential difference between Mg_2_Sn and the matrix. When the Ga content exceeded 0.5 wt %, Mg_5_Ga_2_ was formed and acted as the cathode phase to accelerate the corrosion rate.

Based on the above analysis, at the initial stage of corrosion, the initiation site of Mg–5Sn–0.5Ga alloy is likely to cause corrosion. However, the corrosion initiation sites of Mg–5Sn–1Ga multiply with immersion time, promoting further corrosion. Therefore, in the case of immersion test, the Mg–5Sn–0.5Ga showed superior corrosion resistance than the Mg–5Sn–1Ga alloy.

Precipitation of Mg(OH)_2_, as the main corrosion product on the surface, occurs when Mg^2+^ from anodic dissolution meets OH^−^ from water reduction [37]. The XRD patterns (Figure 9) of the investigated alloys after immersion for 24 h evidence the presence of Mg(OH)_2_ on the surfaces, and the peak intensities of Mg(OH)_2_ reflections decrease with increasing Ga content. This indicates that the corrosion resistance of the Mg–5Sn–*x*Ga alloys was improved by the addition of Ga. As the addition amount of Ga was very small, the corrosion products containing Ga were not detected by XRD.

Figure 10 shows the BSE micrographs of the Mg–5Sn–*x*Ga alloys in cross-section after immersion in 3.5 wt % NaCl solution for 72 h. The samples have good uniformity. After many experiments, it was found that the corrosion depth of all three samples with the same composition showed little discrepancy. As given in Figure 10, corrosion started from the matrix near the second phases and penetrated the matrix along the area of the element-rich region. The thickness of the corrosion layer was the result of measuring the deepest corrosion in a sample. The inserts are an enlarged view of the deepest corrosion of the samples. Again, the thickness of the surface film decreases with increasing Ga content, indicating the positive effect of Ga on influencing the corrosion resistance of the alloys. Figure 10a reveals that the corrosion layer of the Mg–5Sn alloy was 99.1 μm in thickness. The mismatch between the matrix and corrosion product would induce local mechanical stress. Such stress increases with the gradual increasing thickness of the film after immersion [46], and once it reaches a critical value, the surface film will break. The aggressive solution will penetrate the substrate surface through the broken membrane. The relatively complete surface film can protect the substrate to some extent. As a result, the pits grow along a vertical direction.

According to the immersion test, the Mg–5Sn–0.5Ga alloy is more corrosion-resistant than the Mg–5Sn–1Ga alloy. However, this is not consistent with morphologic observations made in the cross-section. The surface film of the Mg–5Sn–0.5Ga alloy (54.7 μm) is thicker than that of the Mg–5Sn–1Ga (38.8 μm) alloy. The calculated mass loss estimates the average corrosion rate in a whole immersion period, while the corrosion section reflects the local corrosion level. It is significant to see that local corrosion occurs around the second phase along the interdendritic region to the interior of the matrix in Mg–5Sn alloy (Figure 10a). The only second phase in the Mg–5Sn–0.5Ga alloy is Mg_2_Sn; hence, the two alloys should behave following the same corrosion mode. However, the oxidation state of Ga is higher than Mg in the Mg(OH)_2_ lattice, which locally induces positive charge in the brucite lattice. Such an increase in positive charge is apt to be balanced by the interaction between Ga element and the harmful Cl^−^ anions, slowing down the penetration of Cl^−^ into the hydroxide layer [15].

To sum up, the second phases, Mg_2_Sn and Mg_5_Ga_2_, exhibit more positive corrosion potentials than the α-Mg matrix [42]. They act as the cathode, and α-Mg the anode, resulting in galvanic corrosion in 3.5 wt % NaCl solution. It is worth noting that Mg_5_Ga_2_ phase has a volta potential difference (VPD) vs. matrix around +75 mV, which will cause marginal local corrosion if the immersion time is not long [19]. Therefore, the presence of Mg_5_Ga_2_ phase increases the area ratio between cathode and anode, which is not conducive to the corrosion resistance, but the local corrosion sensitivity does not increase. Nevertheless, a problem with Ga addition is the uneconomic cost of metallic gallium. Therefore, higher addition levels of Ga content were not performed in this study.

## 4. Conclusions

The main findings of this work are:The microstructures of the Mg–5Sn–*x*Ga alloys similarly present a typical dendritic morphology, regardless of the content of Ga. Ga refines the grain structure adequately. An average grain size of 155.9 ± 8.6 μm was obtained for the Mg–5Sn–3Ga alloy, recording a 48.3% reduction in grain size as compared to the Mg–5Sn alloy;When 0.5 wt % Ga is added to Mg–5Sn alloy, no new phase is formed. When Ga content exceeds 0.5 wt %, a new eutectic phase, identified as Mg_5_Ga_2_, is found in the interdendritic region. Increasing the Ga content decreases the area fraction of Mg_2_Sn phase, while gradually increasing that of Mg_5_Ga_2_ phase;Immersion test in 3.5 wt % NaCl solution shows that the corrosion of the studied Mg–Sn–Ga alloys is initiated in a pitting mode, which rapidly propagates with intense H_2_ evolution on the surface. With the addition of 3 wt % Ga, the overall corrosion rate is decreased significantly. Despite the fact that corrosion is also initiated in a pitting mode, the number of pits is reduced, and the propagation rate is decelerated. The potentiodynamic polarization tests are basically in accordance with the immersion tests.

## Figures and Tables

**Figure 1 materials-12-03686-f001:**
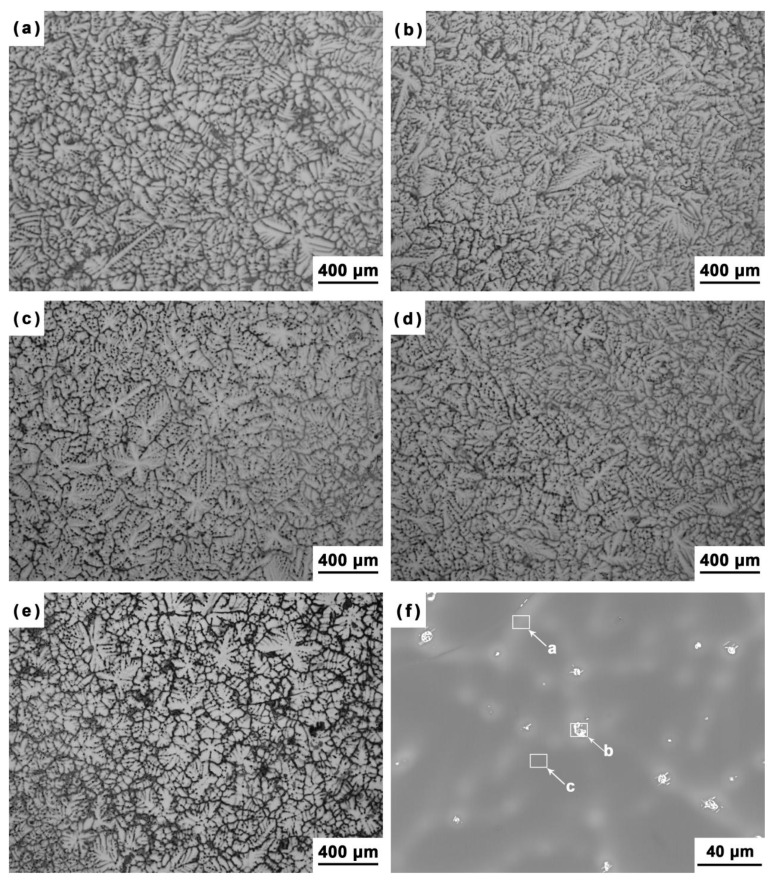
OM micrographs of the Mg–5Sn–*x*Ga alloys: (**a**) Mg–5Sn, (**b**) Mg–5Sn–0.5Ga, (**c**) Mg–5Sn–1Ga, (**d**) Mg–5Sn–2Ga, (**e**) Mg–5Sn–3Ga, and (**f**) backscattered electron (BSE) micrograph of the Mg–5Sn–0.5Ga alloy.

**Figure 2 materials-12-03686-f002:**
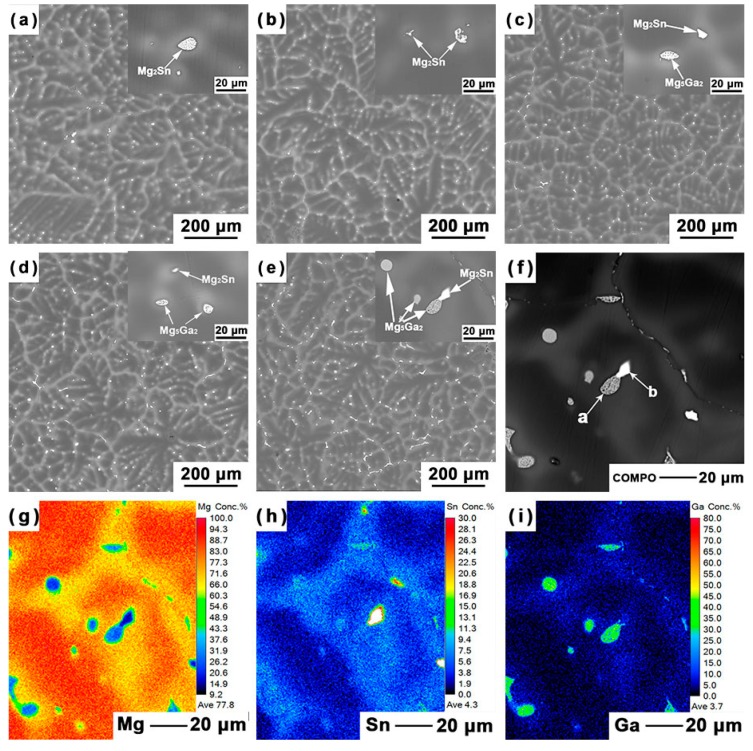
BSE micrographs of the Mg–5Sn–*x*Ga alloys: (**a**) Mg–5Sn, (**b**) Mg–5Sn–0.5Ga, (**c**) Mg–5Sn–1Ga, (**d**) Mg–5Sn–2Ga, (**e**) Mg–5Sn–3Ga. The inserts are zoom-in views showing the second phases. (**f**–**i**) Elemental mapping of the Mg, Sn, and Ga in the Mg–5Sn–3Ga sample using electron microprobe analyzer (EPMA).

**Figure 3 materials-12-03686-f003:**
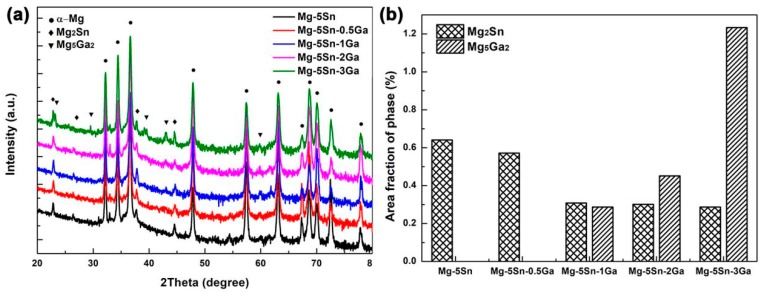
(**a**) X-ray diffraction (XRD) patterns of the Mg–5Sn–*x*Ga alloys; (**b**) the area fraction of the second phases in the investigated alloy.

**Figure 4 materials-12-03686-f004:**
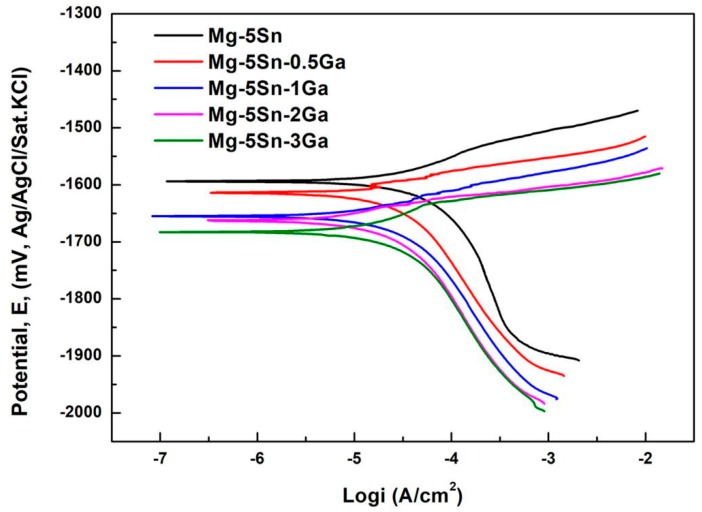
Polarization curves of the Mg–5Sn–*x*Ga alloys in 3.5 wt % NaCl solution.

**Figure 5 materials-12-03686-f005:**
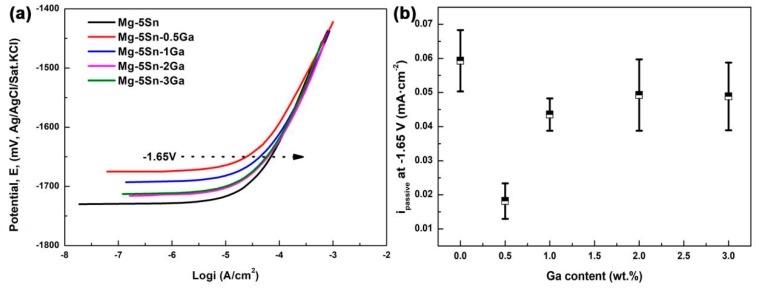
(**a**) The representative polarization curves of the Mg–5Sn–*x*Ga alloys in 0.01 M NaCl solution. (**b**) Passive current density (*i*_passive_) values measured at −1.65 V based on (**a**). The average and standard deviation were obtained from at least five measurements.

**Figure 6 materials-12-03686-f006:**
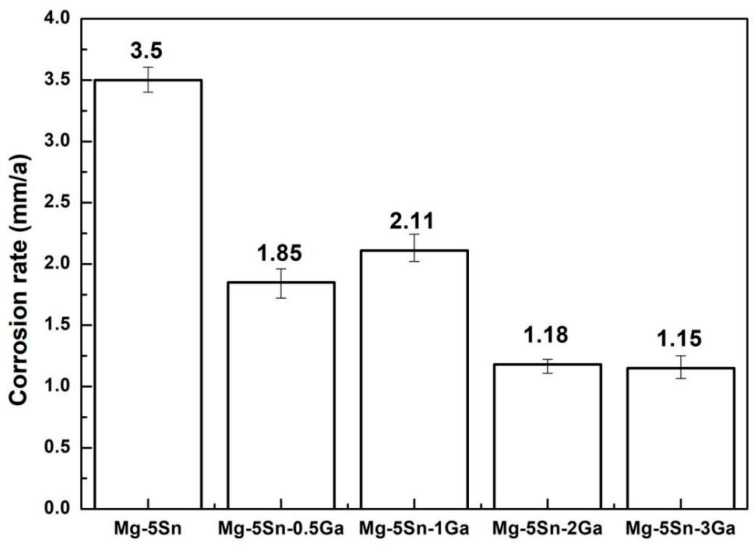
Average corrosion rate of the Mg–5Sn–*x*Ga alloys after immersion in 3.5 wt % NaCl solution for 72 h.

**Figure 7 materials-12-03686-f007:**
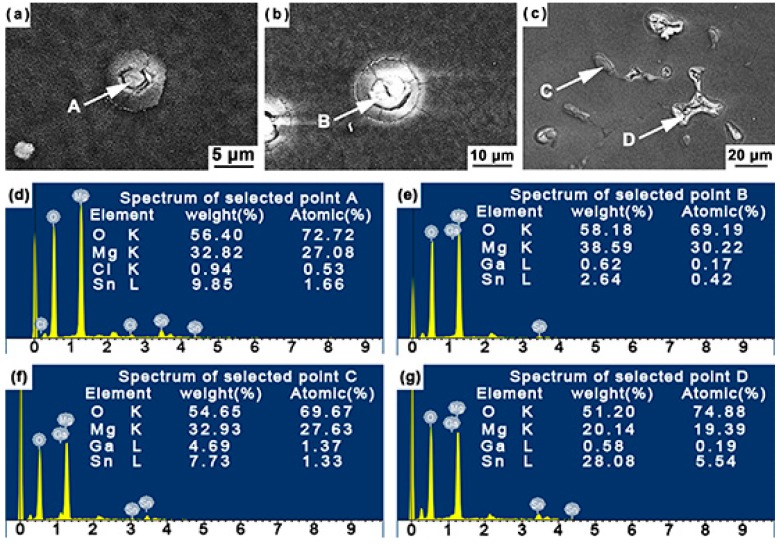
High magnification BSE micrographs showing the alloy surfaces after immersion in 3.5 wt % NaCl solution at 298 K for 6 h: (**a**) Mg–5Sn, (**b**) Mg–5Sn–0.5Ga, (**c**) Mg–5Sn–3Ga; (**d**–**g**) energy-dispersive X-ray spectroscopy (EDS) spectra of the points as indicated by the arrows in the BSE micrographs.

**Figure 8 materials-12-03686-f008:**
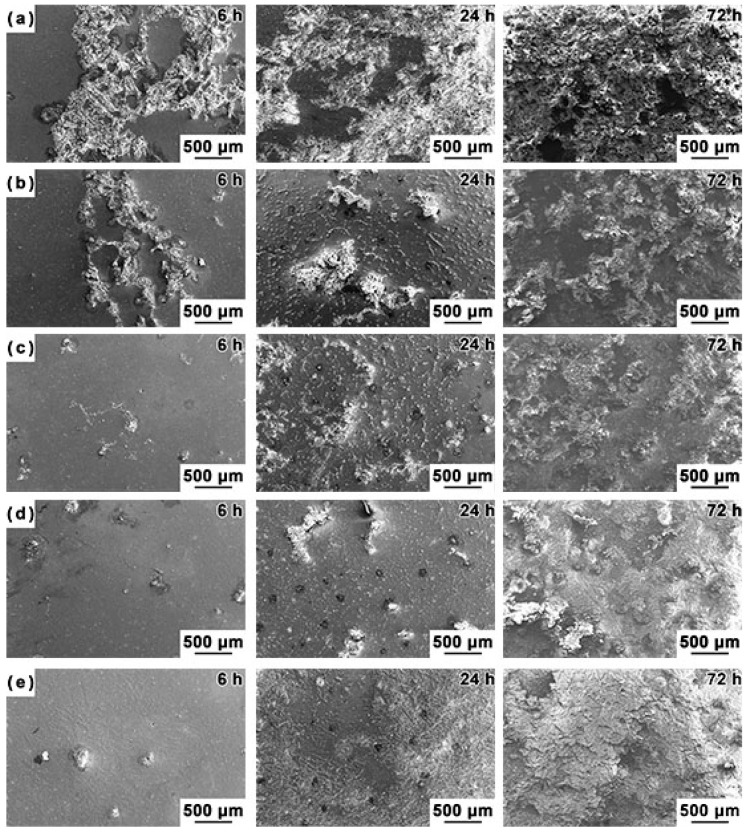
The evolution of the surface morphology of the investigated alloys after soaking for different durations in 3.5 wt % NaCl solution: (**a**) Mg–5Sn, (**b**) Mg–5Sn–0.5Ga, (**c**) Mg–5Sn–1Ga, (**d**) Mg–5Sn–2Ga, and (**e**) Mg–5Sn–3Ga, respectively.

**Figure 9 materials-12-03686-f009:**
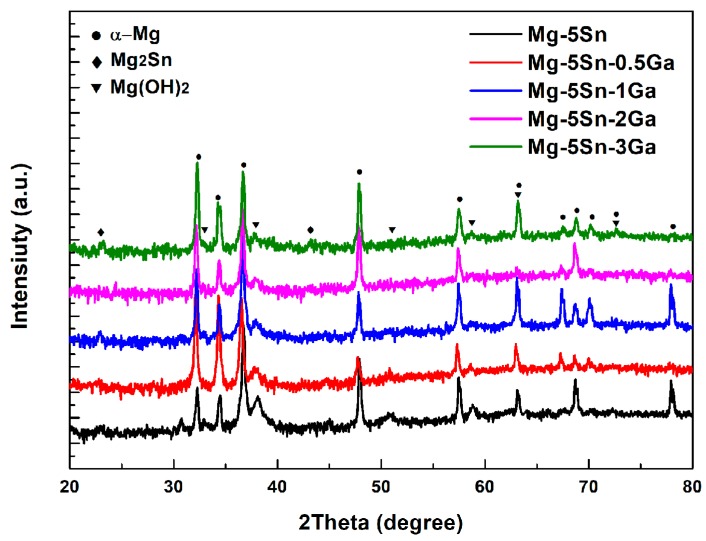
XRD patterns of the Mg–5Sn–*x*Ga alloys after immersion in 3.5 wt % NaCl solution for 24 h.

**Figure 10 materials-12-03686-f010:**
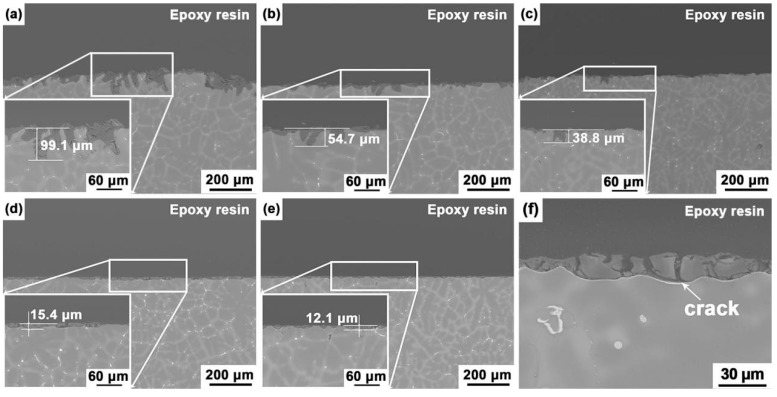
Cross-section morphologies of the Mg–5Sn–*x*Ga alloys after immersion in 3.5 wt % NaCl solution for 72 h: (**a**) Mg–5Sn, (**b**) Mg–5Sn–0.5Ga, (**c**) Mg–5Sn–1Ga, (**d**) Mg–5Sn–2Ga, (**e**) Mg–5Sn–3Ga; the inserts are high-magnification scanning electron microscope (SEM) micrographs of the alloys; (**f**) a typical SEM micrograph showing the cracks on the surface of the Mg–5Sn–3Ga alloy.

**Table 1 materials-12-03686-t001:** Chemical compositions of the experimental Mg–5Sn–*x*Ga (*x* = 0, 0.5, 1, 2, 3 wt %) alloys analyzed by XRF tests.

Nominal Composition	Element Content (wt %)		
	Sn	Ga	Mg
Mg–5Sn	5.08	-	Bal.
Mg–5Sn–0.5Ga	4.98	0.56	Bal.
Mg–5Sn–1Ga	4.81	1.12	Bal.
Mg–5Sn–2Ga	5.08	2.02	Bal.
Mg–5Sn–3Ga	4.67	2.81	Bal.

**Table 2 materials-12-03686-t002:** The chemical compositions of the areas marked by the white rectangles in Figure 1f using the wavelength dispersive spectrometer (WDS) (in wt %).

Area	Sn	Ga	Mg
a	9.778	1.929	88.293
b	25.115	0.741	74.144
c	1.589	0.167	98.244

**Table 3 materials-12-03686-t003:** The chemical compositions of the phases measured by WDS as indicated by the arrows in Figure 2f (in at %).

Point	Sn	Ga	Mg
a	2.2	19.6	78.2
b	29.3	4.4	66.3

**Table 4 materials-12-03686-t004:** Fitting results of polarization curves of the Mg–5Sn–*x*Ga alloys with varying Ga content in 3.5 wt % NaCl solution.

Alloy	Mg–5Sn	Mg–5Sn–0.5Ga	Mg–5Sn–1Ga	Mg–5Sn–2Ga	Mg–5Sn–3Ga
*E*_corr_ (V)	−1.596	−1.614	−1.658	−1.665	−1.684
*i*_corr_ (mA/cm^2^)	9.73 × 10^−2^	3.28 × 10^−2^	3.82 × 10^−2^	3.16 × 10^−2^	2.79 × 10^−2^
*R_i_* (mm/y)	2.223	0.770	0.873	0.722	0.638

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
