# Peer review of "Influence of Alloyed Ga on the Microstructure and Corrosion Properties of As-Cast Mg–5Sn Alloys"

_materials, 2019, doi:10.3390/ma12223686_

Round 1

Reviewer 1 Report

This is an interesting paper about the effect of Ga on Mg-5Sn alloy. However, the “corrosion part” should be corrected.

Line 104, (197) “the average corrosion rate (Ri, mm/a)” What is a? period of time (year, month, day,…) ?

Line 192 “The lowest corrosion current density and corrosion potential are 6.27x10-3 mA/cm2…” Due to the cathodic process control: “in the Tafel extrapolation method for measuring the Mg corrosion rate, the corrosion current density, icorr (mA/cm2) is estimated by Tafel extrapolation of the cathodic branch of the polarisation curve…” [20]. Taking this into account, results of corrosion current density (based on figure 4) should be several times higher than results presented in table 4!

Line 201 “The cathodic branches of the polarization curves show that the cathodic current density decreases with increasing Ga content, demonstrating that Ga indeed retards cathodic reaction and thus reduces corrosion rate” so how to explain that cathodic Tafel slope is bigger in the absence of Ga ?

Polarization technique does not give reliable values for Mg alloys corrosion [4]. Polarisation curves can by useful to characterize of cathodic reaction, passive region or assessing the pitting corrosion potential. Unfortunately, anodic polarisation is too deep (fig 4) and does not allow to discuss about anodic reaction. Authors should correct this.

3.5% NaCl is highly aggressive for magnesium alloys. In practice, concentration of chlorides in environment is much lower. The authors should present how addition of Ga improves resistance of passive layer (by anodic polarisation in solutions containing 1M Cl-, 0.1M Cl-, 0.01MCl-, 0.00.M Cl-.....)

Reviewer 2 Report

At lines 16 and 167 authors said about precipitation of Mg5Ga2 phase, but the term precipitation is not good when they said about eutectic transformation. It could be batter to say that Mg5Ga2 formed by eutectic transformation, not precipitated. At line 104 and elsewhere the units of corrosion rate is mm/a. The mm/a is a mm by age? It is better to change to more appropriate form mm/y (mm by year) here and elsewhere in the text. Line 168. What does it means that Mg2Sn is in not eutectic form? Here and on others microstructures Mg2Sn in the form of divorced eutectic phase and forms via eutectic reaction. For example spheroidal graphite in cast iron is also divorced eutectic phase. Line 174. Please change "Figure 3f" to "Figure 2f". It is not fully understood why authors investigated the alloys in as cast condition. The solution heat treatment and aging is good for that type of alloys because in used amounts Ga can fully dissolved in Mg solid solution. Please give explanation about possible reasons of difference in results obtained via polarization and immersion corrosion tests because the difference is near four times. The corrosion layer thickness measurements is based on only one measure that shown on fig.9 or this is mean value from several measurements? Please specify that in the text.

Reviewer 3 Report

The authors did an excellent work presented in their manuscript. The manuscript is interesting and the results and discussions are presented well. This work can be accepted after language writing revision.

1. Moderate English changes required.

Author Response

Thank you for your comments. We have performed language polishing throughout the context. We believe the writing had been adequately enhanced and hope it meets the high standard for publication

Round 2

Reviewer 1 Report

I can agree that “Tafel slope of polarization curve decreased after the addition of Ga indicating that the corrosion current density decreased,...” but it also indicates that cathodic reaction kinetics is increasing, so nature of the lower corrosion current is different

What is the point to compare anodic currents for the same potential (-1.65V) when alloys show difference in corrosion potential ? Authors should pay attention to the difference in anodic slopes obtained for 0.01M NaCl and 0.1 M NaCl ( collapse anodic branches near the potential 1.55 V - breaking of the passive layer ?)
